# Adjuvant Inhaled Corticosteroids in Community-Acquired Pneumonia: A Review Article

**DOI:** 10.3390/medsci9020034

**Published:** 2021-05-23

**Authors:** Faeq R. Kukhon, Emir Festic

**Affiliations:** Division of Pulmonary and Sleep Medicine, Mayo Clinic Florida, Jacksonville, FL 32224, USA; kukhon.faeq@mayo.edu

**Keywords:** community-acquired pneumonia, inhaled corticosteroids, long-acting beta agonists, systemic corticosteroids

## Abstract

Community-acquired pneumonia is still a major cause of morbidity and mortality worldwide. Since the inflammatory response induced by the immune system is often a major contributor to the lung injury, it becomes reasonable to assess the potential benefit of anti-inflammatory agents in treating community-acquired pneumonia. The role of corticosteroids as adjunct anti-inflammatory agents in treating community-acquired pneumonia is still controversial. Several studies have assessed the benefit of their use in patients with community-acquired pneumonia. In most of those studies, the route of corticosteroids administration was systemic. The aim of this article is to provide a concise review of the role of corticosteroids in treating community-acquired pneumonia when administered via inhalational route, with the potential benefit of avoiding systemic side effects of corticosteroids while exerting the same anti-inflammatory effects on the lungs. Conclusion: the use of inhaled corticosteroids may be of benefit in certain patient subsets with community-acquired pneumonia. Further randomized controlled trials are needed for better determination of such patient subsets.

## 1. Introduction

Lower respiratory infections are still amongst the top ten most common causes of death worldwide according to the World Health Organization (WHO) data [1]. In the USA, community-acquired pneumonia (CAP) has an estimated burden of more than 1.5 million annual adult hospitalizations, 10,000 deaths occurring during hospitalization, and around 1–3% of patients hospitalized within a year of hospitalization [2,3]. In a meta-analysis published by Fine et al., the overall mortality of CAP ranged between 13.6% in hospitalized patients and 36.5% in patients admitted to the intensive care unit [4].

Antimicrobial therapy has been the mainstay treatment of CAP [5,6]. However, in addition to the direct pathogen-mediated lung injury, the inflammatory response elicited by the immune system plays a role in the pathogenesis of CAP, making the utilization of anti-inflammatory agents as an adjuvant therapy a considerable approach in treating CAP [7]. As explained later, several studies and clinical trials have shown potential benefits from using systemic corticosteroids (CS), which have potent anti-inflammatory characteristics, in treating CAP [8,9,10].

With the potential benefits of CS use in CAP, it becomes reasonable to assess the role of inhaled CS in treating CAP, with the advantage of avoiding systemic side effects of CS, while exerting the same anti-inflammatory effects on airways and alveoli using inhaled mode of delivery. This review article sheds light on the potential role of inhaled CS in treating CAP.

## 2. Background and Pathophysiology

CAP represents a clinical syndrome in which acute infection of the lung(s) develops in patients who have not been hospitalized recently and have not been exposed to the health care system. Infection is usually bacterial or viral, though the causative microorganism is not identified in almost half of the cases [6].

The innate immune response, initiated by neutrophils, is critical in determining CAP outcome. Though an inadequate immune response can result in life-threatening infection, an excessive response can potentially lead to life-threatening inflammatory injury as well [7,11].

Although neutrophil products, including cytokines, proteases, and reactive oxygen species, are generated to kill the causative microorganisms, they cause damage to host cells as well [11]. The alveolar-capillary barrier damage caused by the inflammatory mediators leads to protein-rich fluid influx into the alveoli causing noncardiogenic pulmonary edema, which is a defining characteristic of acute lung injury (ALI) [11,12]. Moreover, the subsequent inappropriate accumulation of leukocytes and platelets, and the uncontrolled activation of coagulation pathways are important pathophysiologic concepts of ALI [13]. Existing evidence suggests a relationship between the intensity of the inflammatory response indicated by measured blood levels of inflammatory marker such as interleukin (IL)-6, IL-8, IL-10, tumor necrosis factor (TNF)-α in the blood and the severity of pneumonia and subsequent mortality [14,15]. Under certain conditions, severe inflammatory response could lead to septic shock, multiple organ failure, and death [7,16]. Hence, utilizing medications with anti-inflammatory properties becomes a reasonable intervention to be considered [11].

CS are a class of medications that have a broad spectrum of anti-inflammatory effects, and hence are used for different immune-mediated conditions such as asthma and connective tissue diseases [17]. They non-selectively modulate parts of the host defense by inhibiting the expression and action of pro-inflammatory cytokines including IL-1β and TNF-α as well as other immunomodulatory cytokines such as IL-2, IL-3, IL-4, IL-5, IL-6, IL-8, IL-10, IL-12, and granulocyte-macrophage clony-stimulating factor (GM-CSF). Because cytokines work in cascades, CS block expression of the subsequent cytokines as well. Repair phase cytokines such as transforming growth factor (TGF)-α and platelet-derived growth factor (PDGF) are the exceptions for CS downregulatory effect on cytokine expression and in fact, the expression of those cytokines may even be upregulated by CS. This might explain the modest CS dampening action on healing and subsequent fibrotic processes in lung injury in general [18].

CS have been studied in the preclinical area to assess their benefit (if any) in CAP. In a study performed on piglets, Sibila and colleagues showed that receiving antibiotics along with CS decreased the concentration of IL-6 along with bacterial burden in the bronchoalveolar lavage and lung tissue [19]. In an in vitro experiment, Meduri et al. provided evidence that exogenous CS may restore the host’s ability to counteract infections [20].

Several clinical studies have looked at the potential role of systemic CS in CAP with mixed results [21]. For instance, a randomized controlled trial (RCT) by Nafae and colleagues showed that adjunctive 7-day course low-dose hydrocortisone hastens clinical recovery and prevents the development of sepsis-related complications with significant reduction in the duration of mechanical ventilation and hospital length of stay [8]. In another multicenter double-blinded RCT by Blum et al. (STEP trial), it was shown that a 7-day course of prednisone in patients with CAP shortens time to clinical stability without increase in complications [9]. An RCT by Torres et al. showed that among patients with severe CAP and high initial inflammatory response (defined by elevated C-reactive protein level), the acute use of methylprednisolone compared to placebo decreased the composite endpoint defined as treatment failure [10].

Horita and colleagues concluded in a systematic review and meta-analysis that the use of adjunctive CS in hospitalized patients with CAP seems to be a preferred strategy [22]. Another systemic review by Siemieniuk et al. showed that systemic CS may reduce CAP mortality by 3% and the need for mechanical ventilation by 5% [23]. Several other studies showed similar benefits from CS in CAP (Table 1) [24,25,26,27,28,29].

Conversely, an RCT by Marik and colleagues showed that giving hydrocortisone prior to antibiotic treatment had no effect on the serum TNF-α or the clinical course of patients with severe CAP [30]. Another RCT showed that a 7-day course of prednisone did not improve mortality in CAP [31]. Salluh et al. concluded in a systematic review published in 2008 that using CS as a standard care in patients with severe CAP is not supported by available studies [32].

## 3. Role of Inhaled CS in CAP

Given dissociation between systemic and pulmonary anti-inflammatory effects of systemic CS [29] and the established anti-inflammatory effects of inhaled CS in the alveolar compartment, *inhaled* CS are attractive candidates for adjuvant therapy in patients with pneumonia. Moreover, *systematic* CS have been associated with adverse effects, including hyperglycemia, increased rate of bleeding and secondary infections [21]. Such side effects can be avoided or significantly minimized with the use of *inhaled* CS. There is a decent body of literature that supports, at the very least, clinical equipoise and dictates need for further trials.

### 3.1. Preclinical Studies

Several experiments performed on animal models have shown potential benefit from inhaled CS in ALI, whether given prior to lung injury or after. Forsgren et al. showed in their experiment on pigs with induced *Escherichia coli* endotoxemia that pretreatment with aerosolized CS liposome counteracted the impairment in the dynamic respiratory compliance, expiratory resistance and mean pulmonary artery pressure, with no restrictive influence on the endogenous cortisol production [33].

In another experiment by Jansson et al., pretreatment with inhaled budesonide totally prevented the increased production of TNF-α, IL-1β, IL-6 and monocyte chemo-attractive protein (MCP)-1 after inducing lung injury in rats using lipopolysaccharide. Moreover, pretreatment with budesonide totally prevented lung edema formation and had partial effects on ALI and leukocyte influx [34]. Walther and colleagues evaluated the effect of nebulized beclomethasone on septic pigs after infusing live *Staphylococcus aureus* in an RCT and showed preservation of systemic mean arterial pressure, lung compliance and oxygen delivery with nebulized beclomethasone [35].

In an experiment performed on pigs with induced ALI using chlorine gas, nebulized budesonide was shown to be associated with better oxygen delivery, respiratory compliance and pulmonary vascular resistance [36]. Another experiment performed on pigs by Wang et al. showed that using inhaled budesonide immediately or 30 min after chlorine gas lung injury had similar positive effects on symptoms and signs of pulmonary injury [37].

On the contrary, Sjöblom et al. showed in an animal model that inhaled CS therapy had no effect on ammonia-induced lung injury [38]. This might be explained by the different mechanism of injury produced by ammonia, with more direct and stronger insult on the lung tissues due to its chemical nature, and with a relatively lesser role of inflammation in the subsequent lung injury.

### 3.2. Clinical Studies

Very few clinical studies have assessed the role of inhaled CS alone in treating CAP (Table 2). A retrospective cohort study assessed the association between prehospital use of inhaled CS and the risk of ALI in patients at risk. Though the overall unadjusted incidence of ALI between both groups (inhaled CS group versus non-inhaled CS group) was shown to be nonsignificant, the unadjusted incidence of ALI in patients with ALI due to direct mechanisms: pneumonia (mainly), aspiration, contusion, smoke inhalation or near-drowning, was significantly lower at 4.1% versus 10.6% in patients not on inhaled CS (*p* = 0.006). After propensity matching, the estimated effect for ALI in the whole cohort as well as in patients with direct ALI was statistically nonsignificant at 0.69 (*p* = 0.18) and 0.56 (*p* = 0.24), respectively. Due to risk of overmatching, the authors performed a sensitivity analysis by using traditional multivariate logistic regression that adjusted on all variables found to be significantly associated in the primary univariate analysis; inhaled CS use was found to be the only covariate significantly associated with the primary outcome of ALI in the whole cohort along with LIPS (Lung Injury Prediction Score) and APACHE II (Acute Physiology, Age and Chronic Health Evaluation) scores. The authors explained that the study design along with the uncertainty about the reliability of inhaled CS exposure and subsequent misclassification bias might have partially affected the results [39].

Most of the clinical studies assessed the combination use of inhaled CS and long-acting beta agonists (LABA) since these medications are frequently used combined for chronic airway obstruction. There is evidence from animal models and clinical trials that suggests potential added benefits from LABA to inhaled CS. LABA increase beta adrenergic signaling with cAMP formation and subsequently enhance endothelial barriers function and improve alveolar fluid clearance [40,41]. In addition, the synergistic effect is supported by the bronchodilation caused by LABA, which may improve peripheral drug delivery. Moreover, CS increase the number of beta 2-receptors by increasing gene transcription [42].

A retrospective analysis assessed the association between prehospital use of inhaled CS and inhaled beta agonists with the incidence of ARDS. It was shown that prehospital inhaled CS use was significantly associated with lower risk of ARDS development; 10.5% of inhaled CS users versus 14% of patients not on inhaled CS developed ARDS during the hospitalization (odds ratio (OR) 0.72, 0.53–0.97, *p* < 0.03). After stratification based on the diagnosis of pneumonia, it was shown that the unadjusted effect in all patients was mostly due to the effect on patients with pneumonia (OR 0.49, 0.33–0.72, *p* < 0.001) rather than among those without pneumonia (OR 0.93, 0.59–1.48, *p* = 0.77). After adjusting for confounders, use of prehospital inhaled beta agonists was independently protective of ARDS among all patients (OR 0.48, 0.31–0.72, *p* < 0.001), but not inhaled CS use (OR 0.87, 0.57–1.29, *p* = 0.49) [39]. However, about 70% of the patients using inhaled beta agonists used inhaled CS as well, making distinction of the individual drug effects between the two very difficult.

In a double-blinded, multicenter feasibility trial by Festic et al., sixty patients at high risk for acute respiratory distress syndrome (ARDS) were randomized to receive an *early* 5-day course of inhaled budesonide and formoterol versus placebo (median time between presentation and treatment administration was less than 9 h). The primary outcome, oxygenation improvement, assessed by the ratio of pulse oxygen saturation to the fraction of inspired oxygen (S/F) was significantly improved in the treatment group. The improvement in oxygenation was limited to the subgroup of patients with pneumonia (N = 37, preplanned analysis). Results also showed that patients in the treatment arm had lower rates of mechanical ventilation (53% versus 21%, *p* = 0.01) and ARDS (23% vs. 0%, *p* = 0.01), and shorter hospital and ICU length of stays. There was an imbalance in shock between the two groups (14 (47%) in the placebo versus 4 (13%) in the treatment group); however, oxygenation improved in all patients with shock who received inhaled CS/BA compared to only 43% of those who had received placebo [43].

### 3.3. Viral Pneumonia

#### 3.3.1. Influenza

Influenza has been among the most studied viral pathogens causing CAP. In a meta-analysis by Ni et al., a total of ten studies was included. Compared with placebo, *systemic* CS use was associated with higher mortality (relative risk (RR) 1.75, 95% CI: 1.30–2.36, *p* = 0.0002), longer ICU length of stay (mean difference (MD) 2.14, 95% CI: 1.17–3.10, *p* < 0.0001) and a higher rate of secondary infection (RR 1.98, 95% CI: 1.04–3.78, *p* = 0.04). However, none of the included studies was an RCT. Moreover, most of the included studies did not report the CS doses. More importantly, baseline characteristics of the patients varied among the included studies [44].

An updated Cochrane systematic review by Lansbury and colleagues showed increased mortality odds (OR 3.9, 95% CI: 2.31–6.60, I[2] = 68%) associated with CS use in influenza pneumonia. However, all twenty-one studies that were included to assess the association with mortality were observational studies. A high degree of correlation between CS treatment and potential confounders such as disease severity was noted in many of the studies that reported unadjusted effect estimates. Additionally, time to hospitalization, antiviral medication use, presence of respiratory failure prior to CS and the rationale for CS use were sparsely reported across studies. Therefore, the authors considered the quality of evidence to be very low [45]. No studies have assessed the role of *inhaled* CS in treating influenza pneumonia.

#### 3.3.2. Middle East Respiratory Syndrome (MERS)

An observational retrospective study by Arabi and colleagues assessed the association between *systemic* CS and mortality in patients admitted to the ICU for MERS caused by MERS coronavirus (CoV). Results showed that receipt of CS was associated with increased need for invasive ventilation (93.4% versus 76.6%, *p* < 0.0001) and higher 90-day crude mortality (74.2% versus 57.6%, *p* = 0.002). However, after marginal structural modeling, CS use was not significantly associated with 90-day mortality (adjusted OR, 0.75; 95% CI, 0.52–1.07, *p* = 0.12) but was associated with delay in MERS CoV RNA clearance (adjusted HR = 0.35, 95% CI: 0.17–0.72, *p* = 0.005) [46]. The role of *inhaled* CS in treating MERS pneumonia has not been studied yet.

#### 3.3.3. Severe Acute Respiratory Syndrome (SARS)

In a small RCT that included 16 non-ICU patients with SARS caused by SARS-CoV, it was shown that SARS-CoV RNA concentrations in the second and third week of illness were significantly higher in patients who received *systemic* CS early (<7 days from illness onset) versus placebo, with the median time for plasma SARS-CoV to become undetectable being 12 days (11–20 days) versus 8 days (8–15 days), respectively [47]. No studies have looked at *inhaled* CS use in SARS pneumonia.

#### 3.3.4. Respiratory Syncytial Virus (RSV)

The use of CS in RSV infection was investigated in children, with no conclusive evidence of benefit [48]. An observational study on 50 adults hospitalized for RSV infection showed that *systemic* CS use did not affect viral load or shedding in the thirty-three patients (66%) who received CS, with the antibody-blunted response present in patients who received CS [49]. The role of *inhaled* CS in RSV pneumonia has not been studied so far.

#### 3.3.5. Coronavirus Disease 2019 (COVID-19)

The largest RCT that has assessed *systemic* CS use in viral pneumonia is the RECOVERY trial. The trial included 2104 patients with COVID-19 pneumonia caused by SARS-CoV2 assigned to dexamethasone arm versus usual care arm. In total, 22.9% of patients in the dexamethasone arm died within 28 days after randomization versus 25.7% in the usual care group (age-adjusted RR = 0.83, 95% CI: 0.75–0.93, *p* < 0.001). The mortality benefit in the dexamethasone group was restricted to those who were receiving either invasive mechanical ventilation or oxygen support (no details on the level of oxygen support provided), and was not present among those who did not receive oxygen support at randomization [50].

Relative to *inhaled* CS, a recent preclinical experiment showed that ciclesonide suppressed human CoV replication in cultured cells but did not suppress replication of RSV or influenza virus. The findings suggest that the effect of ciclesonide was specific to CoV, proposing its potential candidacy for treating patients with MERS or COVID-19 [51]. A limited number of case reports has been published suggesting potential benefit from ciclesonide in patients with COVID-19 [52,53].

Halpin et al. published a recent review to evaluate whether premorbid use or continued administration of inhaled CS is a potential risk factor for adverse outcomes in acute respiratory infections due to COVID-19, SARS or MERS. Of the fifty-one included studies, none reported any data in regard to prior *inhaled* CS use in patients with SARS, MERS or COVID-19. Moreover, a limited number of studies reported the prevalence of major comorbidities, including chronic respiratory disease, so it was difficult to draw conclusions about the use of inhaled CS in CoV-related respiratory syndromes [54].

In another review article by Pinna and colleagues, mometasone, ciclesonide and budesonide were reported to have in vitro activity against SARS-CoV2. Therefore, authors suggested that the ideal time for administrating *inhaled* CS is the early replicative phase rather than the late inflammatory phase [55].

A recent open label RCT by Ramakrishnan and colleagues compared the use of *inhaled* budesonide versus usual care in adult patients with mild COVID-19 within 7 days of symptom onset. The primary outcome was urgent care visit, emergency department assessment or hospitalization related to COVID-19, and it occurred in 10 patients in the usual care arm versus 1 patient in the budesonide arm (proportion difference of 0.131, *p* = 0.004), with the number needed to treat with inhaled budesonide being 8. The median number of days until clinical recovery was 7 days in the budesonide arm versus 8 days in the usual care arm (*p* = 0.007). The authors concluded that early administration of inhaled budesonide reduced the need for urgent medical care and the time until recovery following early COVID-19 infection [56].

On the contrary, a large observational study by Schultze et al. did not show a role of *inhaled* CS in protecting against death related to COVID-19 in patients with asthma or chronic obstructive pulmonary disease (COPD) [57]. However, the baseline differences between the compared groups, the increased risk of non-COVID-19 death in COPD patients receiving *inhaled* CS and the potential unadjusted confounders likely contributed to the final results [58].

### 3.4. Aspiration Pneumonia/Pneumonitis

As of yet, no RCTs have evaluated the role of CS in the management of aspiration pneumonia/pneumonitis, regardless of the route of administration [59]. Studies that looked at CS use in aspiration pneumonia/pneumonitis did not show significant difference in complications or outcome [60]. Wolfe and colleagues showed in a case-control study that patients who received CS following aspiration subsequently had more Gram-negative bacteria [61]. Therefore, the routine use of CS in aspiration pneumonia/pneumonitis is not endorsed.

However, despite the lack of evidence to support the use of CS in adults with aspiration pneumonia/pneumonitis, the use of *inhaled* budesonide in treating neonates with meconium-aspiration syndrome (which is very similar in mechanism to aspiration pneumonia-pneumonitis) was shown to be beneficial [62,63,64].

## 4. Discussion

The use of CS in managing CAP is still controversial. As shown in this review article, the lack of more definitive RCTs on the role of CS (systemic or inhaled) is a major contributor to the ongoing controversy.

Hence, the most recent American Thoracic Society CAP guidelines published in 2019 recommends against *routine* use of *systemic* CS in adults with non-severe CAP, and suggests that CS are not *routinely* used in adults with severe CAP (including influenza), while endorsing the use of CS in severe CAP with refractory septic shock per Surviving Sepsis Campaign [5].

Results from some trials and observational studies suggest that using CS in certain subsets of patients with CAP could be of benefit. As mentioned earlier, there is strong evidence showing benefit (including mortality benefit) from using CS in selected patients with COVID-19 pneumonia [45,50]. In a study by Endeman and colleagues, the levels of pro-inflammatory cytokines (IL-1 receptor antagonist, IL-6, IL-8 and IL-10) were found to be influenced by the nature of the causative microorganisms, with pneumococcal pneumonia being significantly associated with higher levels. It was also shown in the same study that the decrease in the levels of the studied cytokines was independently influenced by starting CS therapy [65]. A secondary analysis of the STEP trial by Urwyler et al. showed that the effect of CS was stronger in patients with elevated levels of pro-inflammatory cytokines (IL-6, IL-8 and MCP-1) compared to patients with low levels [66].

Further determination of patient subsets may improve with adjuvant CS (systemic or inhaled), which requires designing trials that address the causative agent, disease severity, timing of therapy and patients’ characteristics (host-directed approach) [67].

Trials with features of adaptive and platform designs are currently assessing multiple interventions in CAP management, including use of *systemic* CS (NCT02735707) (REMAP-CAP trial) [68]. The Arrest Respiratory Failure due to Pneumonia (ARREST Pneumonia) trial is an ongoing double-blinded RCT that assesses the efficacy of combined *inhaled* CS and a beta agonist versus placebo for the prevention of acute respiratory failure in hospitalized patients with severe pneumonia (NCT04193878) [69].

## 5. Conclusions

The use of CS, including inhaled CS, may be of benefit in certain patient subsets with CAP. Determining which modes of CS therapy should be administered and in which patients requires more precise RCTs, systematically addressing both etiology and host characteristics.

## Figures and Tables

**Table 1 medsci-09-00034-t001:** Randomized controlled trials assessed systemic corticosteroids in treating community-acquired pneumonia. CAP: community-acquired pneumonia, CRP: C-reactive protein, MODS: multiple organ dysfunction syndrome, TNF-α: tumor necrosis factor α.

Study	Population	Intervention	Major Outcomes
*Nafae* et al. (2013)	Patients with CAP admitted to ICU.	Intravenous hydrocortisone for 7 days.	Intervention group: reduction in the inflammatory markers, duration of mechanical ventilation, duration of antibiotic treatment, pneumonia complications, length of hospital stay and improved oxygenation.
*Blum* et al. (2015)	Patients hospitalized for CAP.	Prednisone daily for 7 days.	Intervention group: shorter median time to clinical stability and higher incidence of in-hospital hyperglycemia. Pneumonia-associated complications were similar in both groups at day 30.
*Torres* et al. (2015)	Patients with severe CAP and CRP level >150 mg/L.	Intravenous methylprednisolone for 5 days within 36 h of hospital admission.	Intervention group: less treatment failure, which was defined as development of shock, need for invasive mechanical ventilation or death within 72 h of treatment. No difference in in-hospital mortality between both groups.
*Confalonieri* et al. (2005)	Patients with severe CAP admitted to ICU.	Intravenous hydrocortisone infusion for 7 days.	Intervention group: significant improvement in the oxygenation and chest radiograph score; significant reduction in CRP levels, MODS score, hospital length of stay, mortality; and delayed septic shock (by day 8 of study).
*Fernández-Serrano* et al. (2003)	Patients hospitalized for CAP.	Intravenous methylprednisolone for 9 days.	Intervention group: improved oxygenation, faster fever improvement and greater radiological improvement by day 7. No statistically significant difference in mortality or need for mechanical ventilation.
*Meijvis* et al. (2011)	Patients hospitalized for CAP.	Intravenous dexamethasone for 4 days.	Intervention group: shorter length of stay and higher incidence of hyperglycemia. No difference in in-hospital mortality or severe adverse events.
*Snijders* et al. (2010)	Patients hospitalized for CAP.	Oral prednisolone for 7 days.	Intervention group: faster defervescence and faster decline in CRP levels. No difference in clinical outcomes in patients with severe CAP (subgroup analysis). No difference in adverse events between both groups.
*Marik* et al. (1993)	Patients with severe CAP.	A single dose of IV hydrocortisone 30 min prior to starting antibiotic therapy.	No significant difference in TNF-α levels between both groups.

**Table 2 medsci-09-00034-t002:** Randomized controlled trials assessed inhaled corticosteroids in treating community-acquired pneumonia. CAP: community-acquired pneumonia, COVID-19: coronavirus disease 2019.

Study	Population	Intervention	Major Outcomes
*Festic* et al. (2017)	Patients admitted through emergency department at risk for ARDS.	Aerosolized budesonide/formoterol for 5 days.	Intervention group: lower rates of mechanical ventilation and ARDS, and shorter hospital and ICU length of stays. Oxygenation improvement was limited to the subgroup of patients with pneumonia.
*Ramakrishnan* et al. (2021)	Patients with mild COVID-19 within 7 days of symptoms onset.	Inhaled budesonide.	Intervention group: less need for urgent care visit, emergency department assessment or hospitalization related to COVID, and shorter time to clinical recovery by 1 day.

## Data Availability

Data sharing not applicable.

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
