# Peer review of "Adjuvant Inhaled Corticosteroids in Community-Acquired Pneumonia: A Review Article"

_medsci, 2021, doi:10.3390/medsci9020034_

Round 1

Reviewer 1 Report

I would like recommand to authors  a large argumentation in physiology and pneumonia pathophysiology in cytokines profile of inflammation. Urwyler et al. 2011 described the association of pro inflammatory cytokines IL 6, IL 8 a MCP-1.  The role of cortikosteroids was decribed in low importance in this study. Endeman et al 2011 analysed the next two cytokines IL-1RA. IL 10. The role of cortikosteroids was described in both papers like the reducing of pro inflammatory cytokines. It´s a good knowlidge for decision making in steroids indication not for Covid19 pneumonia only. 

Author Response

Thank you so much for the valuable feedback. Your suggestions have been reflected in the revised manuscript. We've expanded in our discussion regarding the pathophysiology and the effect of steroids on cytokines (background and pathophysiology section). The suggested papers have been also cited and their content have been discussed under the discussion section.

Reviewer 2 Report

very useful summary of the state of the art of CAP-treatment, an ongoing  and right now even more important discussion. The paper also emphasizes the need for further data which will allow a more precise treatment.

Adjuvant Inhaled Corticosteroids in Community-Acquired Pneumonia: A Review Article

By R. Kukhon and E. Festic

The review addresses the role of corticosteroids as anti-inflammatory agents in CAP with an emphasis on inhalation therapy. This is a clinically relevant point since side effects may be reduced and therapeutic effects on the lung could be improved given the fact that CAP may lead to sepsis, organ failure and death. Focusing on inhalative administration of corticosteroids in CAP, therefore, addresses a relevant and timely topic although corticoid adjuvant therapy in pneumonia sepsis is discussed for at least 15 years.

The paper is comprehensive and nicely written; however, minor points can be improved:

Although including most of relevant papers is hardly possible, some papers should be cited in addition, e.g.:

Salluh JI, Póvoa P, Soares M, Castro-Faria-Neto HC, Bozza FA, Bozza PT. The role of corticosteroids in severe community-acquired pneumonia: a systematic review [published correction appears in Crit Care. 2008 Nov 7;12(6):434]. Crit Care. 2008;12(3):R76. doi:10.1186/cc6922 and
Corticosteroids in Community-Acquired Pneumonia. JAMA. 2020;323(9):887–888. doi:10.1001/jama.2020.0216

 Although a recent paper addressing inhaled corticosteroids  in COVID-19 has been cited (#51, Halpin et al.), and other references discussing the effect of inhaled ciclesonide, the paper would benefit from going a little bit deeper into  the potential relevance of corticosteroid inhalation in Covid-19 treatment, e.g. Pinna SM, Scabini S, Corcione S, Lupia T, De Rosa FG. COVID-19 pneumonia: do not leave the corticosteroids behind!. Future Microbiol. 2021;16:317-322. doi:10.2217/fmb-2020-0199 and
Schultze A, et al. Risk of COVID-19-related death among patients with chronic obstructive pulmonary disease or asthma prescribed inhaled corticosteroids: an observational cohort study using the OpenSAFELY platform. Lancet Respir Med 2020; 8: 1106–1120. And as a commentary to that:
Does inhaled corticosteroid use affect the risk of COVID-19-related death?Alexander Jordan, Pradeesh Sivapalan, Jens-Ulrik Jensen, Breathe 2021 17: 200275; DOI: 10.1183/20734735.0275-2020

Further, the discussion may include an estimation of potential side effects comparing systemic to local inhalative administration routes.

Author Response

Thank you so much for the useful suggestions and valuable comments. All suggested papers have been cited and their content has been reflected in the revised manuscript under background and pathophysiology section and COVID-19 section respectively. We also described the major side effects associated with systemic steroids compared to inhaled steroids.

Reviewer 3 Report

This manuscript focuses on the role of inhaled corticosteroids in community-acquired pneumonia. I believe this is an interesting article and asks important questions; however, there are some major points:

1) please prepare graphical abstract that will appear alongside the text abstract in the Table of Contents. Please remember that it should summarize the content, but also represent the topic of the article in an attention-grabbing way.

2) references list should be supplemented with newer positions

3) in Tables - please provide years of publications stated as references

4) there is a formatting issue in Table 1, row 1: some lines are highlighted

Author Response

Thank you very much for the feedback. Below are point-by-point responses to the comments:

  1. Graphical abstract has been added (under the abstract section).
  2. References have been updated.
  3. Years of publications have been added. 
  4. The formatting issue has been fixed.

Round 2

Reviewer 3 Report

The manuscript has been improved and deserves, in my opinion, to be published.